# Lower Light Intensities Increase Shoot Germination with Improved Leaf Biosynthesis in Ma Bamboo (*Dendrocalamus latiflorus* Munro)

**Lili Fan** [1,2], **Bingjun Li** [1], **Yongzhen Han** [1], **Liguang Chen** [1], **Tianyou He** [3], **Yushan Zheng** [1,3] and **Jundong Rong** [1,*]

1   College of Forestry, Fujian Agriculture and Forestry University, Fuzhou 350002, China
2   Research Institute of Subtropical Forestry, Chinese Academy of Forestry, Hangzhou 311400, China
3   College of Landscape Architecture, Fujian Agriculture and Forestry University, Fuzhou 350002, China
*   Correspondence: rongjd@126.com

**Abstract:** Ma bamboo (*Dendrocalamus latiflorus* Munro) is a major bamboo species cultivated in southern China with high economic, ecological, and social value. However, highly dense forests and reasonable structures in bamboo forests have unclear and adverse effects on light transmittance and forest productivity that are not adequately understood. Here, we investigated varied light-intensity treatments during different phases of shoot emergence and development on Ma bamboo shoots. The amount of total chlorophyll, carotenoids, gas exchange indicators, and biosynthetic products were also compared to explore the response mechanism of shoot germination on downstream biochemical pathways. We found that compared to the L0 treatment (full sunlight), the number of germinated bamboo shoots under the L1 treatment (40% light) increased significantly by 44.07% and 101.32% in the shooting initial-phase and metaphase, respectively ($p < 0.05$). Additionally, the net photosynthetic rate ($P_n$) during the shooting initial-phase and metaphase was the highest in the L1 treatment, while the L4 (10% light) and L0 treatments inhibited chlorophyll synthesis. Further, the accumulation of leaf carbon (C) and nitrogen (N) was higher in the L1 treatment than in other treatments. Ma bamboo showed rich carbohydrate contents under L0 and L1 treatments in the shooting initial-phase and metaphase. Principal component analysis (PCA) also revealed that the L1 treatment positively correlated with bamboo shoot germination and biochemical activity during the shooting periods. Ultimately, our data suggest that the L1 treatment is the most optimal for promoting bamboo shoot germination, providing a scientific basis for cultivating shoot-used bamboo forests in southern China.

**Keywords:** gradient light environment; bamboo shoot germination; photosynthetic characteristics; photosynthetic products; endogenous hormones

## 1. Introduction

In forests, light is a dynamic resource that changes over hours, days, weeks, or months depending on the time of day and seasonal changes [1]. Besides temperature and moisture, light is also an important component that affects the survival, growth, regeneration, and final productivity of understory seedlings and saplings [2–4]. There are significant spatial and temporal differences in light intensity between plant canopies, understory, or gaps, and most plants are shaded to some extent throughout their life cycle. Plant growth requires an optimum photosynthetic flux density (PPFD). A very high or low PPFD can affect plant photosynthesis, ultimately impacting plant productivity and limiting plant growth and development [5]. Therefore, reasonable stand structures and understory light condition improvements are crucial to enhance the productivity of plantation forests.

Light transmittance is essential for plant growth, depending on the species and habitat. In the dual-purpose bamboo forest, appropriate light transmittance is conducive to the development of bamboo shoots and underground stems [5]. Further, the increase of woodland shading that creates lower light intensities can inhibit the reproductive growth

of *Fargesia entrico* and prolong its vegetative growth. However, others have also found that the number of bamboo shoots decreases with increased forest transmittance [6]. High light transmittance is vital for forming the early shoots of *Bambuas entricose* and impacts the number of shoots and Hsinchu in a cluster, which increases with the increase of light transmittance [7]. With *Indocalamus decorus*, full sunlight is crucial to promote the maximum number of lateral bud germination, while shading causes these characteristics to decrease [8]. Ultimately, light can improve the germination rate of bamboo shoots by stimulating shoot differentiation, which is beneficial to advance bamboo forest management and improve their productivity.

Ma bamboo is one of the most critical bamboo species in the world and is mainly cultivated in southern China. It is a large "clump" style bamboo from the Poaceae family, with extremely high economic, ecological, and social value. Compared with scattered bamboo, tufted bamboo is an integrated and independent system, and its nutrients are transferred between reactants to improve resource utilization efficiency [9]. However, this physiological integration is usually affected by the density of a bamboo forest, where the higher the density of the forest, the lower the effect [4,10]. Further, an increase in stand density will severely limit the light transmittance in the woods [11,12]. Chen et al. [7,13] have shown that light can stimulate the germination of *B. oldhamii* shoots, and the number of shoots in a cluster increases significantly with increased light transmittance in a forest. Additionally, a reasonable bamboo forest density structure significantly improves the forest's lighting conditions, stimulates shoot germination, and increases yield.

The southern part of China has sufficient rainfall and suitable temperatures in the late spring and early summer. However, increased rainfall during early bamboo shoot development can severely affect forest productivity [14]. During the shooting stage, the light under the forest becomes an important environmental factor that affects the germination of the bamboo shoots. In the early stages of shoot formation, the rainy weather aggravates natural light resources, and the low light environment seriously inhibits the normal germination of the bamboo shoots, eventually diminishing the bamboo shoots' production [15,16]. However, little research has been performed with Ma bamboo to understand shoot emergence and the response mechanisms of bamboo leaves to different light gradients.

This study aimed to investigate different light intensities with artificial shading to simulate the lighting environment of Ma bamboo forests under natural operating conditions to understand the variables responsible for their productivity. We assigned sufficient natural light (100% light) to the lowest understory PPFD (10% light) as the simulated intensity gradients. Additionally, the response mechanisms of bud germination and biological plasticity to different light intensities were explored, and the physiological response strategy of bamboo shoot germination was also revealed. Further, we characterized the number of germinated bamboo shoots and the biological characteristics of the leaves to changes in light intensity at each shooting stage, which included their photosynthetic capacity, mineral nutrients, carbohydrate storage, and endogenous hormones. Ultimately, this work provides a theoretical basis for adjusting the stand density of Ma bamboo and the efficient cultivation of bamboo shoots that will increase bamboo forest productivity in southern China.

## 2. Materials and Methods

### 2.1. Plant Materials

Three-year-old Ma bamboo seedlings were used that had an average seedling height of 105.51 cm, an average ground diameter of 4.66 mm, and an average crown width of 71.11 cm (north–south) and 68.95 cm (east–west). Seedlings were planted in a non-woven bag with red soil. The weight of the potted substrate was 15.99 kg, and the substrate pH was 5.77. The organic carbon content was 13.67 g·kg$^{-1}$, and the total nitrogen, total phosphorus, and total potassium contents were 0.35, 0.50, and 50.01 g·kg$^{-1}$, respectively. The pot experiments were conducted at Fujian Agriculture and Forestry University (119°14′47.37″ E, 26°05′29.88″ N).

*2.2. Experimental Design*

The pot experiments used a completely randomized experimental design, and the shading treatment was carried out on one day (10 April 2019). A total of five light intensities were set in the experiment: 100%, 40%, 30%, 20%, and 10%, which were denoted as L0, L1, L2, L3, and L4, respectively. Each treatment contained eight replicates. Shading sheds (3 m in height, 3 m in length, and 6 m in width) were built with steel frames and black plastic shade nets. Each shed was separated by 1.20 m to reduce mutual interference between treatments.

The setting of the light intensity gradient was based on natural management conditions. The bamboo forests that showed differences in the number of stands due to different land types (i.e., forest land, farmland, and riparian alluvial land) were selected as the survey objects. The Taiwan Hipoint portable and handheld spectrometer (HP350) was used to measure simultaneous understory illuminance and PPFD stage changes. With this, the ratio of PPFD under the forest and the bare ground was used to convert the light intensity. The measurement time was 6:00–18:00, and measurements were collected every 2 h interval. Measurements were collected every 15 d starting in February 2019 in four consecutive cycles to comprehensively obtain the light intensity changes under different common management types in Ma bamboo forests (Figure A1). The measurement results showed that the variation range of the light intensity under the Ma bamboo forests was 10%–40%. Shading was used to simulate lighting under the forest in the pot experiments. The average PPFD that corresponded to 8:00 and 15:50 (Table A1) was selected. The PPFD value, using different numbers of needles and layers of plastic shading nets, was measured to obtain the corresponding light intensity.

After one month of light treatments, the light-response curve of the leaves was determined, and the light saturation point (LSP) was obtained. In the shooting initial-phase (mid-June 2019), shooting metaphase (early August 2019), and shooting anaphase (early October 2019), the number of shoot buds in each period was counted, and the physiological indicators of the leaves were determined. During the experiments, the soil moisture content was kept above 60%, and 10 g of compound fertilizer (N:P:K = 15:15:15) was applied to each cluster in May. The maximum temperature in the shed was $\leq 35\,^\circ$C, and the relative air humidity was maintained at $\geq 85$%. During the test period, management measures such as weeding, insecticide, and pruning were also performed.

*2.3. Measurement Indicators and Methods*

2.3.1. Investigation of the Germination Number for Bamboo Shoots

Without destroying the soil structure, the number of germinated shoots was recorded after the shoot tips were unearthed (Figure A2), and the tip length was usually $\leq 5$ cm [17]. At the beginning of each treatment, the number of shoots in each shooting stage was counted every day. In the shooting initial-phase and metaphase, the germinating new shoots were cut off from the base after they had grown to a height of approximately 50 cm, and the new bamboo shoots were retained in the shooting anaphase ($n = 8$ per treatment).

2.3.2. Determination of Leaf Photosynthetic Characteristics

Photosynthetic Pigments

Each pot's upper, middle, and lower mature leaves were selected as mixed samples, and pigments were extracted by the direct extraction method, as stated in Gao [18]. The optical densities of the extracts were measured at 645, 663, and 470 nm by UV spectrophotometer (TU-1901, Beijing Puxi General Instrument Co., Ltd., Beijing, China), and the total chlorophyll content (Chls) and carotenoids (Car) were calculated according to the Lichtenthaler method ($n = 4$ per treatment) [19].

Gas Exchange Parameters

Mature functional leaves were selected from the apex of each clump for gas exchange determination. Photosynthetic characteristics were measured using the portable photosyn-

thesis instrument (LI-6400 XT, LI-COR Biosciences, Lincoln, NE, USA) with red and blue light sources and a light saturation intensity of 1600 μmol·m$^{-2}$·s$^{-1}$ PPFD. Before the assay, the leaves were placed under the 1600 μmol·m$^{-2}$·s$^{-1}$ PPFD light intensity for 20–30 min. The measurement indicators include net photosynthetic rate ($P_n$), stomatal conductance ($g_s$), intercellular $CO_2$ concentration ($C_i$), and transpiration rate ($T_r$) ($n = 4$ per treatment).

### 2.3.3. Determination of Carbon (C), Nitrogen (N), and Non-Structural Carbohydrates (NSCs) Contents in Leaves

Each pot's upper, middle, and lower mature leaves were selected in mixed samples ($n = 4$ per treatment). Fresh samples were used to determine sucrose, starch, and soluble sugar contents. All indicators were measured using kits produced by Suzhou Keming Biotechnology Co., Ltd., Suzhou, China. The NSCs content was the sum of soluble sugar and starch contents. Dry samples were used to determine total N and C contents using a carbon-nitrogen elemental analyzer (Vario Max, Elementar, Langenselbold, Germany). The carbon-nitrogen ratio (C/N) was calculated based on the carbon-nitrogen results.

### 2.3.4. Determination of Endogenous Hormone Contents in Leaves

Each cluster's upper, middle, and lower mature leaves were selected from different standing bamboo for mixed samples ($n = 4$ per treatment). The extraction and separation of indole acetic acid (IAA), zeatin (ZT), gibberellins (GA$_3$), and abscisic acid (ABA) were carried out using the improved method from Li et al. [20]. After sample purification, high-performance liquid chromatography (Shimadzu LCMS-2010, HPLC, Kyoto, Japan) was used to determine if IAA, ABA, ZT, and GA$_3$ were present.

### 2.4. Data Analysis

Data analysis was performed using Excel 2016 and SPSS 22.0 software. One-way ANOVA was used to test the significance of each shooting stage ($\alpha = 0.05$), and principal component analysis (PCA) was used to analyze the relationship between the number of germinated bamboo shoots and the biochemical characteristics of the leaves. Prism v. 8.0.1 (GraphPad, San Diego, CA, USA) and Origin 9.5 (OriginLab OriginPro 2019) were used for charting.

## 3. Result

### 3.1. Effects of Light Intensity on the Number of Germinated Shoots for Ma Bamboo

To investigate the effect of different light intensities on germinated bamboo shoots, we analyzed the number of Ma bamboo shoots in each germination phase. Figure 1 describes the number of germinated shoots for each light treatment group (L0 to L4) in the initial-phase, metaphase, and anaphase. The L1 treatment had the highest number of germinated shoots at each shooting stage. In the shooting initial-phase, the number of germinated shoots in the L2 and L3 treatments increased by 35.59% and 26.32% ($p < 0.05$), respectively, while the metaphase also increased by 8.48% and 8.67% ($p < 0.05$), compared to the L0 treatment. However, the number of germinated shoots in the L2 and L3 treatments decreased by 37.78% and 38.89%, respectively, compared with the L0 treatment in the anaphase. Compared with other treatments, the L4 treatment inhibited the shoot germination and was not significantly different from L0 at each shooting stage.

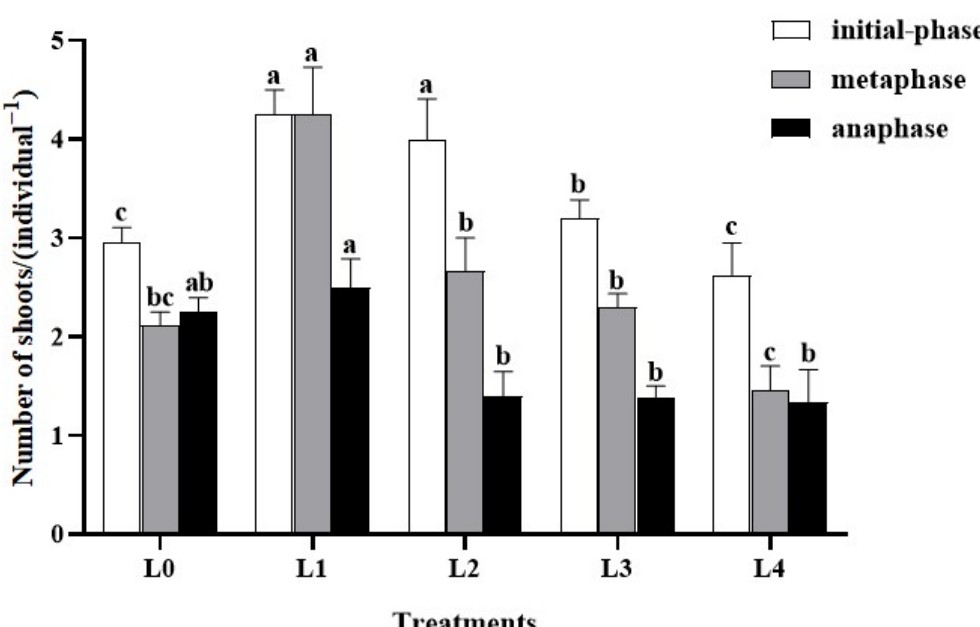

**Figure 1.** Effects of different light intensities on the number of germinated shoots for Ma bamboo. Note: L0, L1, L2, L3, and L4 refer to 100%, 40%, 30%, 20%, and 10% of natural light, respectively. Values are the means ± SE of eight replicates per treatment. Different letters indicate significant differences between treatments at the same shooting period ($p < 0.05$).

### 3.2. *Effects of Light Intensity on Leaf Photosynthetic Properties*

3.2.1. Effects of Light Intensity on Leaf Photosynthetic Pigments

To understand the effect of the different light intensities on biosynthesis pathways, such as photosynthesis, we extracted total chlorophyll (Chls) and carotenoids (Car) from leaves. We analyzed their amounts among the different treatments and varying shooting stages, which is illustrated in Figure 2. Compared with the L0 treatment, the contents of Chls and Car in the shading treatments (except the L3 treatment) increased at the shooting stages. The amounts of Chls and Car with each treatment increased in the shooting anaphase compared to the shooting initial-phase and metaphase. Additionally, the quantities of Chls and Car in the L2 and L3 treatments were significantly higher in the shooting anaphase than those under the L0 treatment ($p < 0.05$).

3.2.2. Effects of Light Intensity on Leaf Gas Exchange Indicators

After investigating light intensity effects on photosynthetic pigments, we analyzed leaf gas exchange indicators to understand the productivity of the Ma bamboo in different light conditions. Figure 3 illustrates the different gas exchange indicators for net photosynthetic rate ($P_n$, A), stomatal conductance ($g_s$, B), intercellular $CO_2$ concentration ($C_i$, C), and transpiration rate ($T_r$, D). In the shooting initial-phase and metaphase, $P_n$ increased with the L1 treatment but decreased with the decrease of light intensity with treatments L2–L4. With the shooting anaphase, $P_n$ under the L0 treatment was significantly higher than in other treatments ($p < 0.05$). Under different light intensities, there were significant differences in leaf $g_s$ in the shooting initial-phase and metaphase ($p < 0.05$), and there were also substantial differences between L0 and L1 treatments in the shooting anaphase ($p < 0.05$). With $C_i$, light intensities also significantly affected the shooting initial-phase ($p < 0.05$). Ultimately, these variable trends for $g_s$ and $T_r$ were the same under different light intensities, and the various treatments had significant effects on $T_r$ in the shooting metaphase and anaphase ($p < 0.05$).

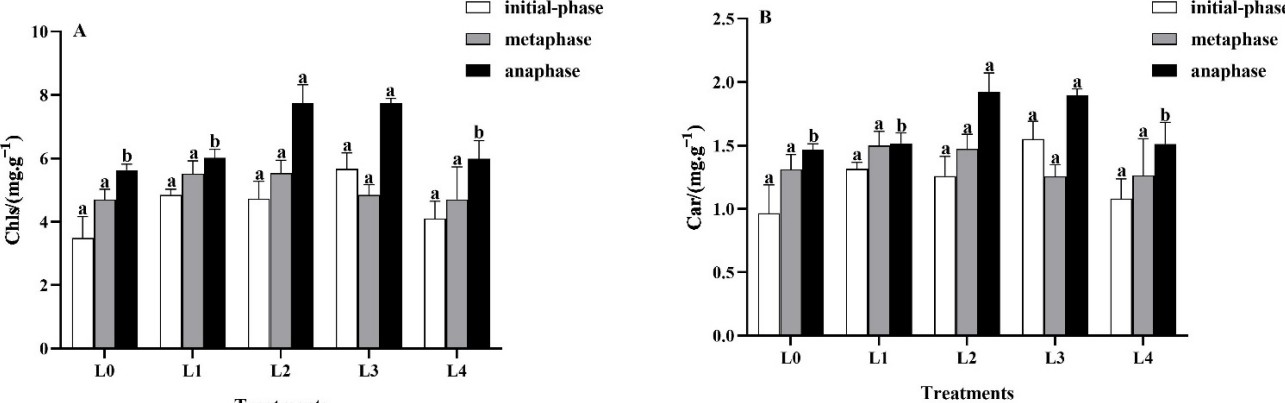

**Figure 2.** Effects of different light intensities on the contents of total chlorophyll (Chls, (**A**)) and carotenoids (Car, (**B**)) in Ma bamboo. Note: L0, L1, L2, L3 and L4 refer to 100%, 40%, 30%, 20%, and 10% of natural light, respectively. Values are the means ± SE of four replicates per treatment. Different letters indicate significant differences between treatments at the same shooting period ($p < 0.05$).

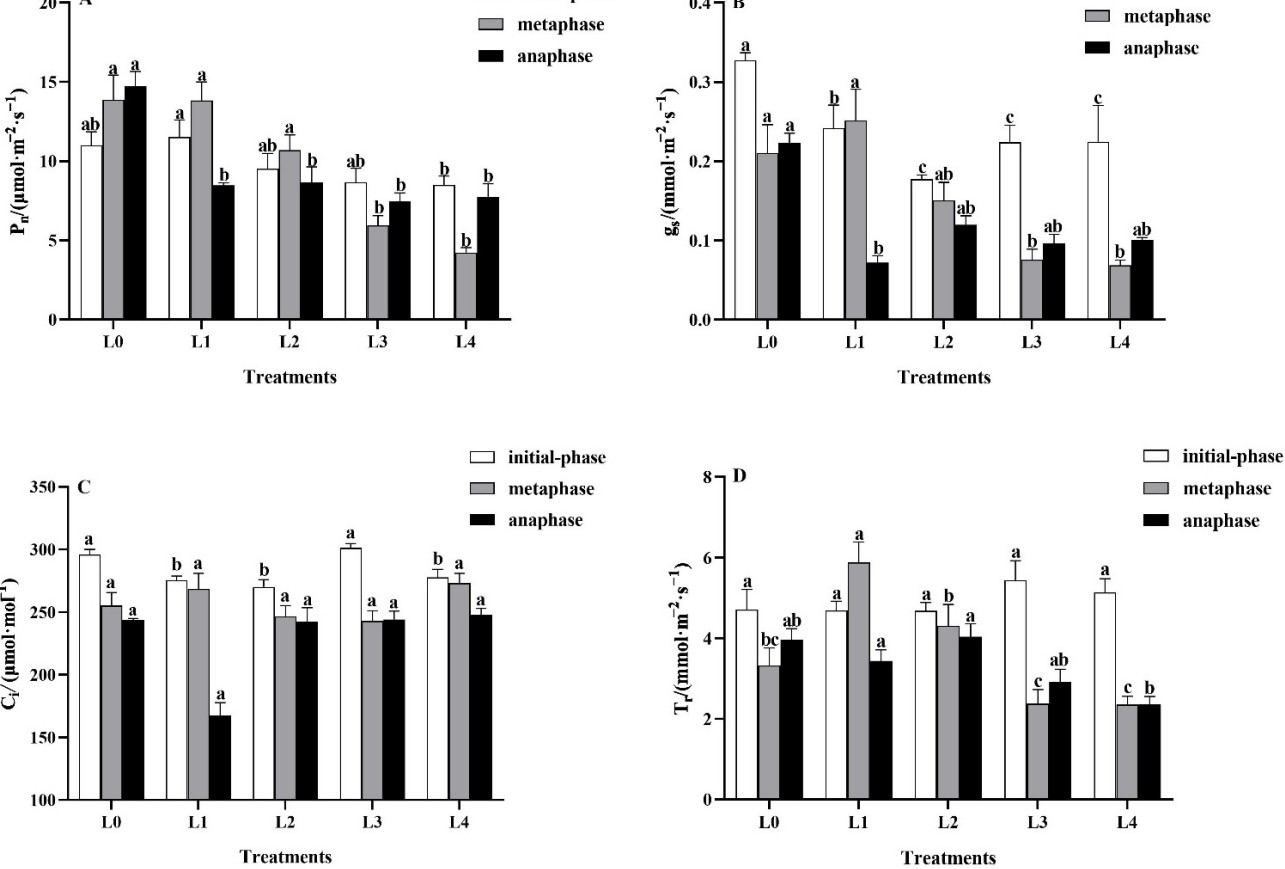

**Figure 3.** Effects of different light intensities on leaf gas exchange indicators for (**A**) net photosynthetic rate ($P_n$), (**B**) stomatal conductance ($g_s$), (**C**), intercellular $CO_2$ concentration ($C_i$), and (**D**) transpiration rate ($T_r$) of Ma bamboo. Note: L0, L1, L2, L3, and L4 refer to 100%, 40%, 30%, 20%, and 10% of natural light, respectively. Values are the means ± SE of four replicates per treatment. Different letters indicate significant differences between treatments at the same shooting period ($p < 0.05$).

### 3.3. Effects of Light Intensity on Leaf C and N Properties

After investigating gas exchange indicators, we also explored light intensity effects on carbon and nitrogen in the Ma bamboo leaves to understand its impact on downstream chemical processes, which are illustrated in Figure 4A–C. In the shooting initial-phase, the contents of C and N were significantly increased under L1, L2, and L3 groups compared with L0 ($p < 0.05$). However, shading treatments significantly decreased N and C contents in the shooting metaphase and antaphase compared with L0 ($p < 0.05$). L4 treatment can highly inhibit the accumulation of C and N at each shooting period ($p < 0.05$). In the shooting initial-phase, each treatment's C and N contents were lower than those in the shooting metaphase and anaphase. Interestingly, the C/N of the shading treatments was significantly higher than that of the L0 treatment in the shooting metaphase, which is contrary to the accumulation of C and N ($p < 0.05$).

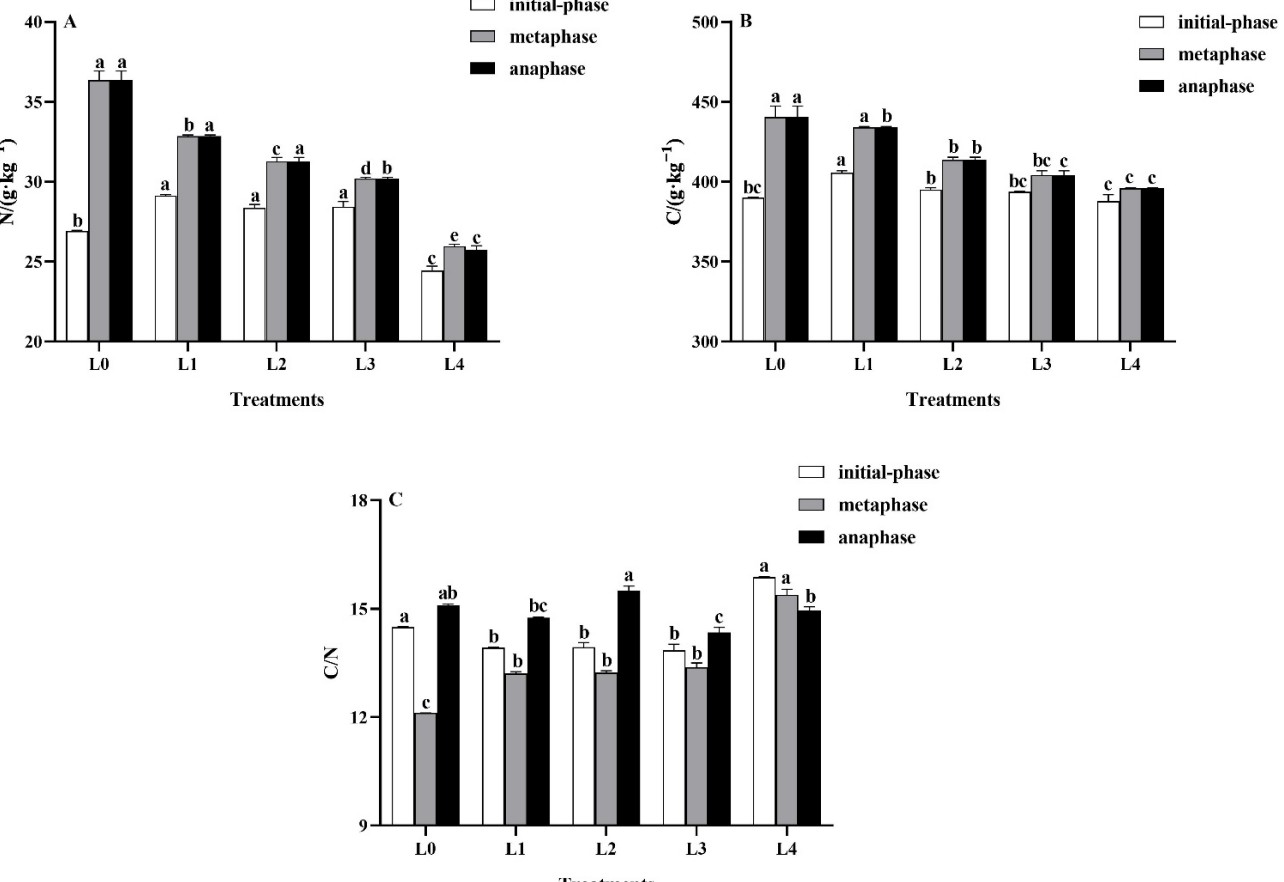

**Figure 4.** Effects of different light intensities on nitrogen (**A**) and carbon (**B**) production and their ratio (**C**) in Ma bamboo. Note: L0, L1, L2, L3, and L4 refer to 100%, 40%, 30%, 20%, and 10% of natural light, respectively. Values are the means ± SE of four replicates per treatment. Different letters indicate significant differences between treatments at the same shooting period ($p < 0.05$).

### 3.4. Effects of Light Intensity on Leaf Carbohydrates

Along with gas exchange indicators and C and N contents, we also investigated the effect of light intensity on leaf carbohydrates. Figure 5A–D displays the amounts of sucrose, soluble sugar, starch, and NSCs in Ma bamboo leaves throughout the different shooting phases and light treatment groups. In the shooting initial-phase, the contents of sucrose, starch, and NSCs in the L1 treatment were higher than those in the L0 treatment. In the shooting metaphase, the shading treatments significantly decreased the leaf sucrose and NSCs ($p < 0.05$), and all carbohydrates showed the lowest values in the L2 treatment compared with L0. At the end of shoot growth, light treatment had no significant effect

on the number of carbohydrates in the leaves, while shading treatments increased starch and NSCs contents compared with L0. Contrary to the shooting metaphase, however, the soluble sugar, starch, and NSCs in the leaves in the shooting anaphase showed the highest amounts with the L2 and L3 treatments.

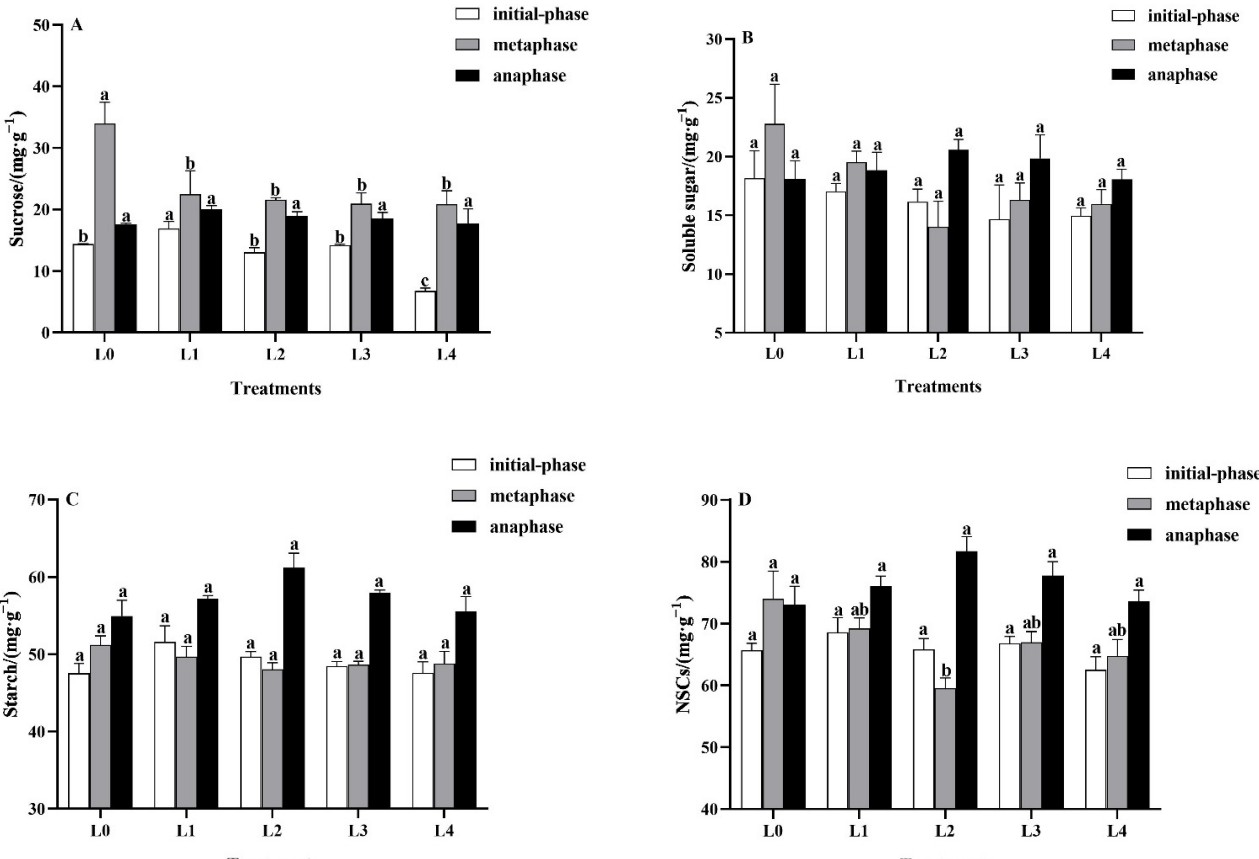

**Figure 5.** Effects of different light intensities on leaf carbohydrate contents of Ma bamboo that include (**A**) sucrose, (**B**) soluble sugar, (**C**) starch, and (**D**) NSCs. Note: L0, L1, L2, L3, and L4 refer to 100%, 40%, 30%, 20%, and 10% of natural light, respectively. Values are the means ± SE of four replicates per treatment. Different letters indicate significant differences between treatments at the same shooting period ($p < 0.05$).

### 3.5. Effects of Light Intensity on Leaf Endogenous Hormones

We further explored downstream biochemical effects with the different light intensities by extracting and analyzing the concentrations of endogenous hormones, which are illustrated in Figure 6A–D. The contents of IAA and $GA_3$ were significantly decreased under the shading treatments compared to the L0 treatment in three shooting periods (both $p < 0.05$). In the shooting initial-phase, the ZT content in the L1 treatment was higher than in the L0 treatment (10.89%), while the ZT in other shading treatments was significantly lower than those in the L0 treatment ($p < 0.05$). Conversely, ABA content in the L0 and L1 treatments was lower than that in other shading treatments. In the middle stage of shoots, the ZT and ABA contents also showed opposite changes to the shooting initial-phase between treatments ($p < 0.05$). In the shooting anaphase, ZT and ABA showed the highest amounts in the L2 and L3 treatments, which was different from the changes in IAA and $GA_3$.

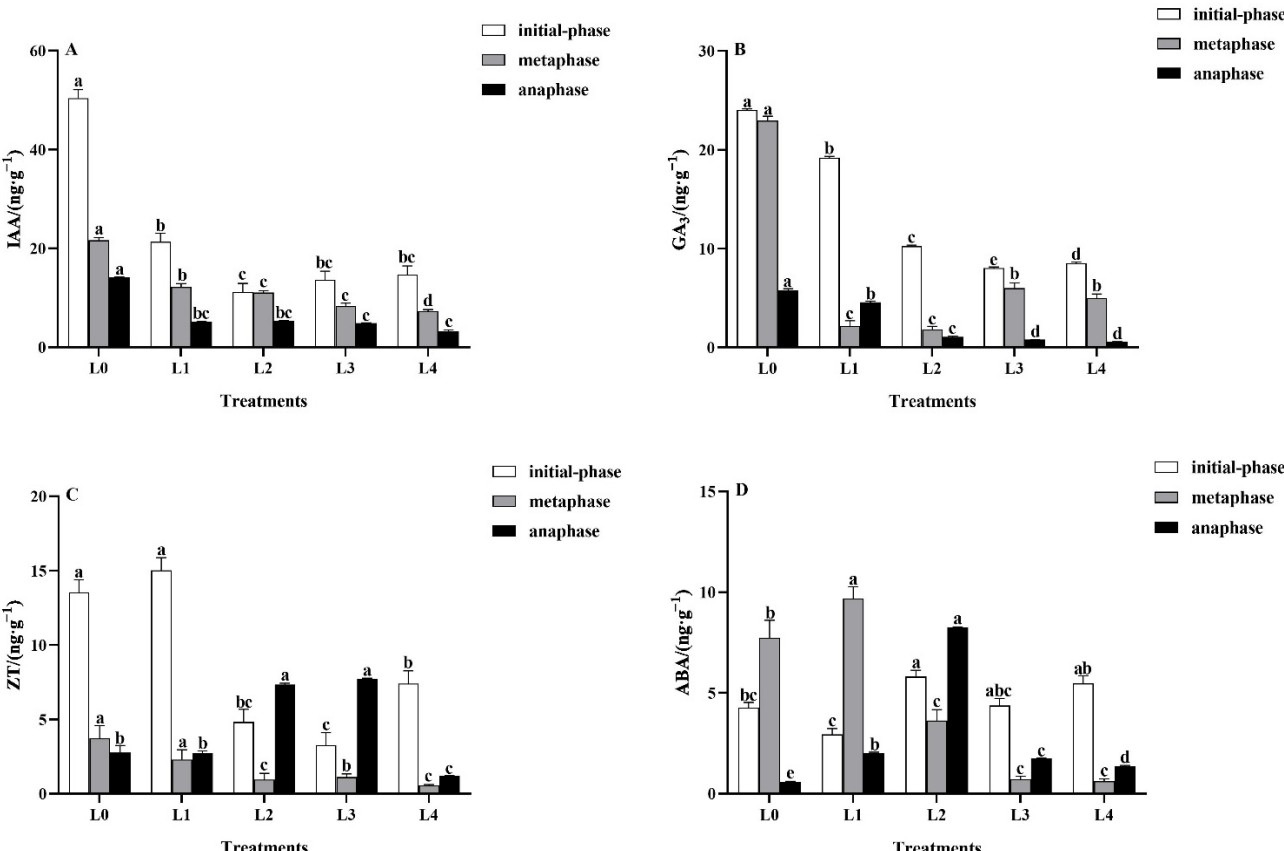

**Figure 6.** Effects of different light intensities on leaf endogenous hormone of Ma bamboo. Hormones include IAA (**A**), GA$_3$ (**B**), ZT (**C**), and ABA (**D**). Note: L0, L1, L2, L3, and L4 refer to 100%, 40%, 30%, 20%, and 10% of natural light, respectively. Values are the means ± SE of four replicates per treatment. Different letters indicate significant differences between treatments at the same shooting period ($p < 0.05$).

### 3.6. PCA of Light Intensity on the Number of Germinated Shoots and Leaf Biochemical Parameters

After investigating the effects of the varying light intensities on gas exchange indicators, pigment characteristics, C and N amounts, carbohydrate characteristics, and endogenous hormones, we used this information to perform PCA analyses to compare the germinated bamboo shoots to these specific biochemical attributes. Figure 7A–C displays these data as a PCA diagram with the number of germinated shoots and all the previous biochemical characteristics described in Figures 1–6. The PCA analysis showed that the cumulative variance contribution rate of the first two principal components was 82.71%, 85.10%, and 66.58%, respectively, in the shooting initial-phase, metaphase, and anaphase; this could explain the variation information of the data.

In the shooting initial-phase, the shoot strongly correlated to Car, Chls, SU, ST, NSCs, C, and N in PC1. Leaf IAA, ZT, and GA$_3$ were highly correlated with SS, P$_n$, and g$_s$, which negatively correlated to ABA. Additionally, the L1, L2, and L3 treatments were strongly related to PC1. In the shooting metaphase, the shoot had strong correlations with Car, Chls, P$_n$, T$_r$, g$_s$, and ABA in PC1. Additionally, leaf carbohydrates highly correlated with IAA, ZT, and GA$_3$. Further, L0 and L1 treatments also had positive correlations with PC1. In the shooting anaphase, the correlation between the shoot and PC2 was higher than PC1. Additionally, the shoot had strong positive correlations with P$_n$ and g$_s$, and negative correlations with leaf photosynthetic pigments, carbohydrates, and endogenous hormones while those indicators were interrelated. Further, the L0 and L1 treatments were strongly correlated with PC2.

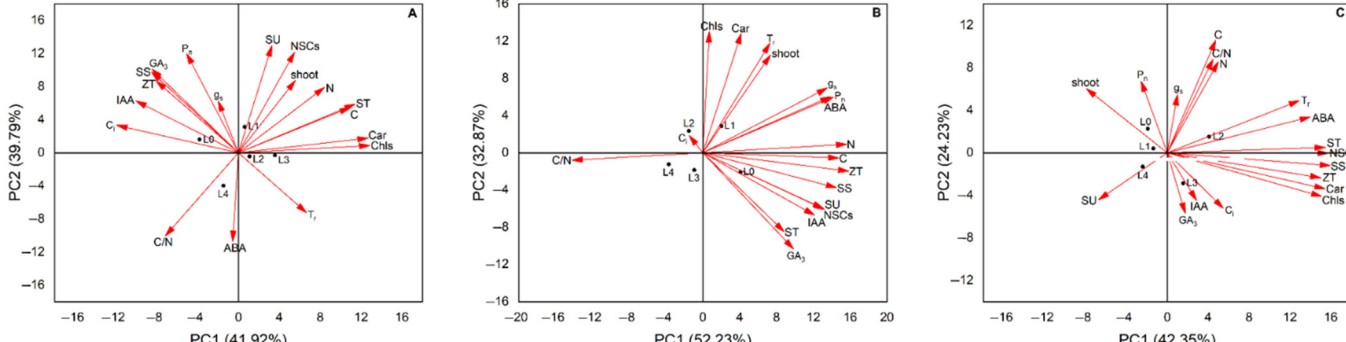

**Figure 7.** PCA diagram of the number of germinated bamboo shoots and leaf physiochemical attributes for Ma bamboo under the different light treatments by shooting period was (**A**) shooting initial-phase, (**B**) shooting metaphase, and (**C**) shooting anaphase. Note: shoot—the number of germinated shoots, Chls—total chlorophyll, Car—carotenoids, $P_n$—net photosynthetic rate, $g_s$—stomatal conductance, $C_i$—intercellular $CO_2$ concentration, $T_r$—transpiration rate, N—total nitrogen, C—total carbon, C/N—total carbon/total nitrogen, SU—sucrose, SS—soluble sugar, ST—starch, NSCs—non-structural carbohydrates, IAA—indole-3-acetic acid, ZT—zeatin, GA$_3$—gibberellins, ABA—abscisic acid; L0, L1, L2, L3, and L4 refer to 100%, 40%, 30%, 20%, and 10% of natural light, respectively.

## 4. Discussion

### 4.1. Photosynthetic Mechanism on Shoot Germination of Ma Bamboo under Light Intensity

Yorio et al. found that plants need to synthesize more chlorophyll to improve photosynthesis efficiency and adapt to weak light environments [21]. Compared with full sunlight, Ma bamboo leaves synthesized more photosynthetic pigments during shooting periods in a shaded environment to improve the leaf—light utilization rate [22]. We found that the number of germinated shoots had a strong positive correlation with Chls and Car synthesis in the initial-phase and metaphase. Ma bamboo accumulated higher Chls and Car concentration with increased shooting capacity from the 20%–40% light. These results indicate that enough Chls is synthesized to use light energy in the weak light environment and ensures plant productivity, consistent with past *Emmenopterys henryi* [23] and *Acer davidii* research [24]. Additionally, the leaves showed more significant potential for carbon assimilation [25] with higher accumulations of photosynthetic pigments in the shooting anaphase. In contrast, Chls, Car, and the number of germinated shoots had weak correlations.

Besides nutrient effects, leaf gas exchange indices showed significant plasticity in response to substantial light [26,27]. Ma bamboo showed a higher $P_n$ under full sunlight during the shoot development. However, the 40% light increased $P_n$ in the shooting initial-phase and metaphase, which had more advantages to enhance the leaf's solid carbon capacity and has been confirmed in previous research [28]. In the metaphase and anaphase, the number of germinated shoots showed a strong positive correlation with $P_n$ and $g_s$. Specifically, the 40% light positively regulated $P_n$ and $g_s$, increased leaf net photosynthetic capacity, enhanced photosynthetic carbon fixation capacity, and provided a material basis for shoot germination. Higher $g_s$ can improve the optical biochemical pathway and promote the accumulation of photosynthesis products [19,29,30], which provides the material foundation for bamboo shoots' germination. In the shooting anaphase, $P_n$ and $g_s$ elicited the highest performance under full sunlight conditions, and the higher PPFD promoted the germination of the lateral bamboo shoots. In the shooting metaphase, the number of shoot germination was positively correlated with $T_r$. Under 40% light, the high $T_r$ value at the shooting stage enabled Ma bamboo to have a higher transpiration rate and water transport capacity. Ma bamboo can maintain a good leaf water status and avoid the effects on shoot germination caused by the weakening of photosynthetic biochemical pathways due to stomata limitation [31,32].

### 4.2. The Effects of Leaf Photosynthesis Products on Shoot Germination of Ma Bamboo under Light Intensity

The concentration of C and N in plants reflects nutrient absorption, utilization efficiency, and adaptation to environmental stress. In a specific range, higher leaf N content can prolong the photosynthesis time, accelerate the photosynthetic rate, and improve the nutrient absorption capacity of plants [33]. In comparison, lower N content may lead to a decrease in Chl a, resulting in reduced Chl a/b and photosynthetic efficiency of leaves [34]. In the shooting initial-phase, the number of germinated shoots had a strong positive correlation with the photosynthetic pigment and C and N content. Under 40% light, the contents of leaf Chls and Car and the value of $P_n$ increased, and the photosynthetic capacity was enhanced, which promoted the accumulation of C and N in leaves. Additionally, the increased leaf N content can put more N into photosynthetic pigments to improve leaf $CO_2$ fixation capacity by enhancing the low light utilization efficiency [35,36]. Consequently, the number of germinated shoots in 40% light was higher than in other treatments for the initial-phase. However, the number of germinated shoots was not strongly correlated with the C-N metabolic indicators in the metaphase.

In the anaphase, C, N, and C/N were important parameters that affected the germination of bamboo shoots. The C and N contents under 40% and 100% light were significantly ($p < 0.05$) higher than other treatments, consistent with the changes in the number of germinated shoots. With the prolongation of the shooting periods, the natural light intensity gradually decreased, and the photosynthesis of plants growing in low light conditions fixed less C and required fewer nutrients than plants growing in high light [37,38], which resulted in insufficient nutrient power for bud germination. The contents of C and N in the shooting anaphase were highly ($p < 0.05$) reduced in 20%–30% light, and the number of germinated shoots decreased.

The shade-tolerant tree species can accumulate higher carbohydrate contents under weak light [39,40]. In the shooting initial-phase, the germination of bamboo shoots had a strong positive correlation with sucrose, starch, and NSCs. Compared to full sunlight, the accumulation of sucrose, starch, and NSCs in the leaf under 40% light provides a material basis for shoot germination [41]. In the shooting metaphase, the reduction of leaf carbohydrates under 10%–40% light may be related to the photosynthetic capacity compared to full sunlight, where Ma bamboo under 40% light can maintain a higher $P_n$, which was more likely to allocate more C to meet other metabolic needs. Therefore, reducing leaf NSCs can meet the needs of shooting in the underground part, to improve the ability of bamboo shoots to germinate, which is consistent with Hu et al. [42]. The accumulation of photosynthetic products in the leaves is a negative feedback mechanism for the accumulation of photosynthesis products [43,44]. In the shooting anaphase, the accumulation of NSCs under 10%–30% light led to decreased $P_n$ in the leaves. The soluble sugar, starch, and NSCs were negatively related to the germination of bamboo shoots in the shooting anaphase. A considerable accumulation of carbohydrates in the leaves and the decrease in $P_n$ would affect lateral bamboo shooting under 10%–30% light. Conversely, the collection of carbohydrates under 40% and 100% light decreased. However, it still maintains a high $P_n$, possibly because the distribution of photosynthesis to bamboo stump ensured the nutritional needs of lateral bamboo shoots [45,46].

### 4.3. The Effects of Leaf Endogenous Hormones on Shoot Germination of Ma Bamboo under Light Intensity

Previous studies have shown that the delayed effect of weak light stress on the tiller of *Gramineae* is complicated by the content of their endogenous hormones [47]. In the shooting initial-phase, the leaves contained hormone concentrations at IAA > $GA_3$ > ZT > ABA, respectively, which indicated that the changes in IAA and ABA were central in responding to different light gradient changes. Specifically, high IAA content was conducive to accumulating photosynthesis products [1]. In the shooting initial-phase, IAA had a strong positive correlation with soluble sugar. High IAA content was conducive

to the accumulation of soluble sugar in leaves under 40% and 100% light. Additionally, Rook et al. found that high IAA levels can relieve the stoma closure caused by ABA accumulation [48]. We also found that IAA had strong positive correlations with $P_n$ and $g_s$. Further, the ABA content was the lowest under 40% light. Therefore, it can enhance photosynthesis by improving the leaf net photosynthetic capacity and promoting the stoma's opening [49] to maintain the high $P_n$ and $g_s$. However, in 10%–30% light, IAA was significantly ($p < 0.05$) lower than that of 40% and 100% light, and ABA showed higher accumulation, which decreased $P_n$, $g_s$, and carbohydrates. In the shooting initial-phase, the endogenous hormone content in the leaves did not have a strong correlation with the number of germinated shoots. These changes in endogenous hormones could affect a plant's photosynthetic ability and photosynthesis products, which will indirectly affect the shoots of Ma bamboo.

In the shooting metaphase, ABA had a strong positive correlation with the number of germinated shoots. The ABA content under 40% light was higher, and the number of germinated shoots was significantly ($p <0.05$) higher than in other treatments, which was different from other works where the accumulation of ABA was negatively related to the tiller [47], possibly because the ABA content enhanced the ability of plants to adapt to low light and relieved the inhibitory effect of shooting due to the high temperature and soil water deficit [50]. Meanwhile, other studies have shown that reducing $GA_3$ content in leaves contributes to the growth of tillers [51,52]. In the shooting metaphase, $GA_3$ content under 30%–40% light significantly decreased ($p < 0.05$), which helped promote shoot development. Additionally, $GA_3$ content can promote the synthesis of IAA, and low $GA_3$ content can indirectly control and reduce IAA content [53], which is consistent with the results of this study. With this, the synergistic effect of IAA and $GA_3$ indirectly affects plant assimilation allocation [52,53]. Further, IAA and $GA_3$ had a strong positive correlation with carbohydrates, and the contents of IAA and $GA_3$ under 30%–40% light were significantly lower than those under full sunlight. However, their interaction could promote the distribution of carbohydrates to the bamboo stump and indirectly affect the germination of the bamboo shoots.

In the shooting anaphase, the changes of IAA and $GA_3$ were consistent with the trend of $P_n$ and $G_s$, both performing highest at 40% and 100% light, which did not negatively regulate the net photosynthetic capacity. However, in 20%–30% light, higher ABA and ZT related closely to abundant carbohydrates, which had a negative feedback mechanism for net photosynthetic capacity. Low ABA and ZT concentrations in 40% and 100% light can improve photosynthesis [54] and photosynthetic carbon fixation capacity, which were conducive to promoting shoot germination [55]. These also fully indicated that ABA and ZT had an important regulatory role in the net photosynthetic ability and the accumulation of photosynthetic products at the end of the shoot.

## 5. Conclusions

In this study, we found that the synthesis and distribution of photosynthesis products by improving the photosynthetic characteristics of bamboo were affected by moderate light intensity. Changing the endogenous hormones to enhance the photosynthetic capacity and photosynthate transport during the bamboo shooting period stimulated the germination of bamboo stumps. The 40% light contained higher Chls and Car, maintained higher $P_n$, and increased photosynthetic C fixation capacity in the leaves, which was beneficial to the accumulation and distribution of leaf carbohydrates. These can help provide nutrients for bamboo shoot emergence throughout the bamboo shoot period. Under 40% light, the contents of IAA and $GA_3$ were relatively higher, which could enhance photosynthetic capacity, promote carbohydrate transfer, and indirectly increase the number of germinated shoots. In the shooting initial-phase and anaphase, the lower ABA concentration in the leaves indirectly enhanced leaf photosynthesis, while in the shooting metaphase, higher ABA improved the physiological status of bamboo at the shooting stage and was beneficial to stimulate shoot germination. Therefore, moderate shading can improve the light

adaptation ability of Ma bamboo seedlings. Ultimately, this study shows that cultivation and productivity can be improved by adjusting the stand structure and maintaining the appropriate light environment and suggests new light intensity information that can be used to strengthen bamboo productivity in other regions in China.

**Author Contributions:** L.F. and J.R. designed this study and improved the English language and grammatical editing. L.F. wrote the first draft of the manuscript and performed the data analysis. B.L. and Y.H. did the fieldwork. L.C., T.H. and Y.Z. gave guidance and methodological advice. All the coauthors contributed to the manuscript's discussion, revision, and improvement. All authors have read and agreed to the published version of the manuscript.

**Funding:** This work was supported by the "National Key R&D Program of China" (2021YFD2200501) and the "Forestry Peak Discipline Construction Project of Fujian Agriculture and Forestry University" (72202200205).

**Data Availability Statement:** The data presented in this study are available in the article.

**Conflicts of Interest:** This is the first submission of this manuscript, and no parts of this manuscript are being considered for publication elsewhere. All authors have read and approved the content of the manuscript. No financial, contractual, or other interest conflicts exist for the study.

**Appendix A**

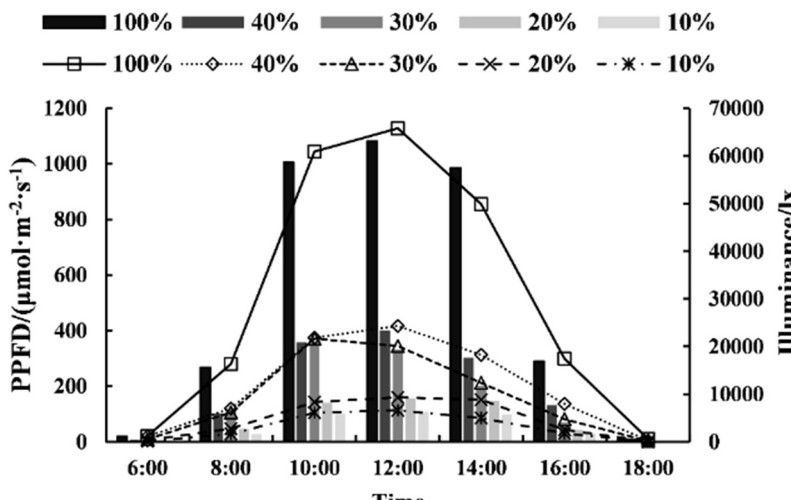

**Figure A1.** Daily variation trend of light intensity under different management types for Ma bamboo. Note: The line graph represents the daily change in PPFD, and the bar graph represents the daily illuminance change. L0, L1, L2, L3, and L4 refer to 100%, 40%, 30%, 20%, and 10% of natural light, respectively.

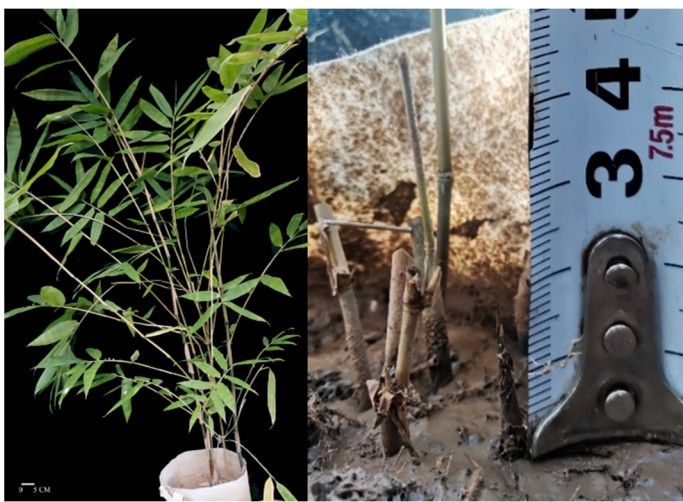

**Figure A2.** The germination of bamboo shoots of Ma bamboo.

**Table A1.** Variation of average light intensity under different management types of Ma bamboo. Note: Data in the same column with different letters denote a significant difference ($p < 0.05$).

| Treatments | Light Intensities (%) | Illuminance (lx) | Average PPFD ($\mu mol \cdot m^{-2} \cdot s^{-1}$) | The PPFD at the Time Corresponds to the Average PPFD ($\mu mol \cdot m^{-2} \cdot s^{-1}$) | |
|---|---|---|---|---|---|
| | | | | 8:00 | 15:50 |
| L0 | 100 | $30491.43 \pm 569.73$ a | $519.62 \pm 64.20$ a | $532.55 \pm 20.37$ a | $545.78 \pm 33.59$ a |
| L1 | 40 | $11394.89 \pm 580.98$ b | $203.89 \pm 10.56$ b | $202.76 \pm 10.97$ b | $206.65 \pm 34.02$ b |
| L2 | 30 | $9070.34 \pm 570.49$ b | $164.57 \pm 9.53$ bc | $177.18 \pm 8.35$ c | $157.03 \pm 10.25$ c |
| L3 | 20 | $6248.78 \pm 503.85$ b | $110.34 \pm 26.06$ c | $79.87 \pm 20.27$ cd | $83.66 \pm 7.39$ cd |
| L4 | 10 | $3129.54 \pm 498.35$ b | $53.64 \pm 13.01$ c | $53.25 \pm 10.04$ d | $54.58 \pm 11.25$ d |

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
