# Peer review of "Lower Light Intensities Increase Shoot Germination with Improved Leaf Biosynthesis in Ma Bamboo (Dendrocalamus latiflorus Munro)"

_forests, doi:10.3390/f13101723_

Round 1

Reviewer 1 Report

In this study, the authors found that the synthesis and distribution of photosynthesis products by improving the photosynthetic characteristics of bamboos was affected by moderate light intensity. Changing the endogenous hormones to enhance the photosynthetic capacity and photosynthate transport of during bamboo shooting period stimulated the germination of bamboo stumps. The authors have provided enough data for the publication and the manuscript was well written. This study will support to improve Ma bamboo cultivation and productivity.

Tables and Figures caption, including supplementary files: All Tables and Figures should be self-explanatory. Please add legends or explain abbreviations to the Tables and Figures captions as needed.

The introduction part is missing some important references.

  https://doi.org/10.1002/fes3.229

https://doi.org/10.1093/plcell/koac193

Reviewer 2 Report

I would like to begin by congratulating the authors on behalf of the thorough and well designed experiments, literature revision in general, statistical analysis and general presentation of the results. I am also convinced of the importance of studies of this nature, that integrate different physiological and biochemical traits, with the purpose of improving plant productivity, both in natural and cultivated environments. Another valuable asset of this study, is that it was dealt with a bamboo that has a great ecological and commercial potential. To date, physiological studies in bamboos are relatively scarce and much needed. This study reflects a well planned design, meticulous work and generated valuable results. However, I have the impression that these results need to be analyzed in more depth. The discussion needs edition and in its present form is perhaps cursory. I strongly suggest the authors to submit this manuscript to a careful edition, with the assistance of a native English speaker, in order to provide an manuscript that may give them full credit for their hard work. Also, proof scientific terms used in plant ecophysiology and physiology and in some cases, check their significance, in order to maintain coherency in the discussion.
